# Research on the expansion, shrinkage properties and fracture evolution of red clay stabilised with phosphogypsum under dry-wet cycles

**Jinxiong Chen, Kaisheng Chen** *, **Zeyu Liu**

School of Civil Engineering, Guizhou University, Guiyang, Guizhou Province, China

* chen_kaisheng@163.com

## Abstract

In view of the special engineering properties of red clay and the waste of phosphogypsum resources, the expansion and contraction deformation and fissure evolution of phosphogypsum stabilized red clay under different conditions were investigated by laboratory tests and image processing system. The research results show that: (1) the absolute expansion and absolute shrinkage of phosphogypsum stabilized red clay are positively correlated with the compaction degree, the number of dry and wet cycles and the cement dosage, and negatively correlated with the initial water content and the phosphogypsum dosage; (2) the fissure rate increases with the increase of the number of dry and wet cycles, and decreases with the increase of the initial water content, the compaction degree, the cement, and the phosphogypsum dosage; (3) The relationship among absolute expansion rate (absolute shrinkage), degree of compaction and fracture rate can be fitted by the equation $f(x,y) = ax +by+cx^2+dy^2+e$; (4) Phosphogypsum has an obvious inhibiting effect on the expansion, shrinkage and cracking of the mix. It is recommended that the cement mixing amount of 6% and phosphogypsum: red clay = 1:1~1:2 as roadbed filler.

## 1 Introductory

Red clay has the characteristics of "high liquid limit (50%-70%), high plasticity index (22%-40%), high natural water content (30%-40%)", which is widely distributed in southern China, and there are a lot of problems when it is used as roadbed filler, such as unstable roadbed, prone to uneven settlement, drying cracking, strength deterioration, etc. At the same time, the poor permeability of red clay can lead to poor drainage problems [1–4]. It is unavoidable to encounter red clay foundations in the actual project, if the replacement method is used to deal with, it will greatly increase the project cost, and the disposal of waste red clay is also a major problem, based on this, most of the current use of cement, fly ash, lime, etc. improved red clay, in order to improve the performance of its road [5–8], which is important for the use of red clay in the construction of the road.

Phosphogypsum is a solid waste produced by industrial production of phosphoric acid. The main components are calcium sulfate dihydrate ($CaSO_4 \cdot 2H_2O$) and $Na_2SiF_6$, while containing

**Funding:** Funding support: Supported by the Provincial Science and Technology Programme of Guizhou Province (Qiankehe Basic-ZK[2023] Key 016; Qiankehe Support[2020] 4Y038). Funders were not involved in study design, data collection and analysis, publication decisions, or manuscript preparation.

**Competing interests:** The authors have declared that no competing interests exist.

a small amount of soluble $P_2O_5$, $SiO_2$, $Al_2O_3$, fluoride, organic matter and other harmful impurities, as well as a small amount of arsenic (As), chromium (Cr), lead (Pb) and other toxic elements, and sometimes residual phosphoric acid, sulphuric acid and hydrofluoric acid [9]. According to the current production process, each ton of phosphoric acid is prepared, which will be accompanied by nearly 5 tons of phosphogypsum. At present, the main treatment measures for phosphogypsum are on-site stacking and land landfill. These traditional treatment methods will bring many negative problems such as environmental pollution and waste of resources [10–13]. It is found that using phosphogypsum for engineering construction can consume a large amount of phosphogypsum and save resources, and the current research on phosphogypsum has made some progress. Zong Wei et al [14] used phosphogypsum to replace part of the fine aggregate, and the mechanical properties of phosphogypsum pavement base material were comparable to those of ordinary cement stabilized gravel through indoor tests. Luo Guofu et al [15] found through consolidation test that the plain red clay is medium compression, and phosphogypsum stabilised red clay is medium and low compression.Qi [16] et al. found that phosphogypsum can be used as an amendment to improve the physicochemical properties of the soil, delay soil degradation, passivation of heavy metals, alternative to the natural gypsum material for the use of resources, and that phosphogypsum can help the soil to capture more carbon, which can help to achieve the carbon emission reduction and carbon neutralisation global goals. Zhou Mingkai [17] et al. explored the influencing factors of the strength properties of cement phosphogypsum stabilized crushed stone, compared with cement stabilized crushed stone, cement phosphogypsum stabilized crushed stone at all ages of the unconfined compressive strength, splitting strength is higher, the growth of the modulus of rebound is slow, it is a kind of good toughness of the road material. Wang Lei [18] found through unconfined compression tests that with the increase of phosphogypsum content, the unconfined compressive strength of the mixed cementitious material decreases.

Liu Chun et al [19] developed a pore (particle) and fissure image recognition and analysis system (PCAS) for studying the micro-formation mechanism of compression-tightening zones in high-porosity sandstones and verified the feasibility and convenience of the method, but its automation is limited. Qi Xiang, Ouyang Miao and other scholars calculated the fracture and porosity of geotechnical bodies with the help of PCAS [20,21], and the method has been widely used for quantitative calculation of fracture and porosity.

It can be seen from the above that there are some research results on the strength and deformation of phosphogypsum stabilized soil, but the previous research is limited to the use of phosphogypsum mixed with other materials (soil, gravel, sand), which plays an auxiliary role instead of being used as the main part. The content of phosphogypsum is low, and due to the particularity of red clay (high liquid limit, high void ratio and high-water content), few scholars have studied the road performance of phosphogypsum stabilized red clay. In this paper, laboratory tests and image processing system (PCAS) were used to study the swelling and shrinkage properties and crack propagation of phosphogypsum stabilized red clay under dry-wet cycles, which provided a theoretical basis for the application of phosphogypsum stabilized red clay materials in road engineering and improved the engineering properties of red clay.

## 2 Raw material properties

The red clay used in the test was taken from K14+000~K16+000 section of Fuquan reconstruction and expansion project, and the depth of the soil taken was 0~3m, the soil sample was reddish-brown, with dense structure, high natural water content, high cohesion and belonged to high liquid limit clay, and its basic parameters are shown in Table 1.

Table 1. Basic physical indexes of red clay.

| $\rho$/g·cm$^{-3}$ | $\omega$/% | $\omega_{op}$/% | $\rho_{dmax}$/g·cm$^{-3}$ | $W_L$/% | $W_P$/% | Cu | Cc |
|---|---|---|---|---|---|---|---|
| 1.76 | 60.03 | 30.24 | 1.46 | 82.13 | 43.02 | 10.63 | 1.085 |

Phosphogypsum was taken from urnfu phosphorus mine in Fuquan City, Guizhou Province, which was dark brown, grey and with large water content. The basic physical index of phosphogypsum is shown in Table 2, the chemical composition is shown in Table 3, and the detection results of heavy metal and radioactivity are shown in Table 4. According to the detection results in Table 4, the heavy metal content of phosphogypsum is in accordance with the relevant provisions of the national standard (GB 5085.3–2007), According to the relevant provisions of GB6566-2010 "Radionuclide Limit for Building Materials", the specific activity of natural radionuclides radium-226, thorium-232 and potassium-40 in the main materials of the building should meet $I_{Ra} \leq 1.0$ and $I_\gamma \leq 1.0$ at the same time, and the measured phosphogypsum has $I_{Ra} = 0.3$ and $I_\gamma = 0.3$, which meets the relevant requirements.

The cement is Southwest brand P.C32.5R silicate cement, grey, dry, no lumps, the basic parameters are shown in Table 5. After testing, all the indexes of the cement sample are in line with the requirements of the technical indexes of General Silicate Cement (GB 175–2007).

The optimum water content and maximum dry density of the mixes with different mixing ratios and different cement dosages were obtained through the compaction test and are shown in Table 6.

## 3 Test scheme

The test procedure is shown in Fig 1.

The tests performed in this paper were conducted with permission, and the laboratory was the Experimental Center of the School of Civil Engineering, Guizhou University.

### 3.1 Sample preparation

Zhang Ying [22] et al. concluded through the 7d unconfined compressive strength test that the unconfined compressive strength of cement-phosphogypsum-red clay was the greatest when the cement dose was 4%~8% and the mass ratio of phosphogypsum to red clay was 1:2.25~1:2.75, according to which the cement admixture of the present test was formulated as 4%, 6% and 8%. In order to analyse the influence of phosphogypsum dosage on the performance of the mixture, the ratio of phosphogypsum and red clay is 1:1, 1:2, 1:3, 1:4, 1:5. According to the "highway roadbed design specification" (JTG D30-2015) [23] for each grade of highway roadbed fill minimum compaction requirements, three and four grade highway compaction is not less than 90%, and one grade highway compaction is not less than 96%, so this paper draws up the degree of compaction of 90%, 92%, 93%, 94%, 95%, and 96%, to analyze the impacts of different degrees of compaction on the performance of the mixture. It is pointed out in the literature [24] that when red clay is used as roadbed fill, the strength and stability of the roadbed soil can be guaranteed within the range of 5% above and below the optimum water content, so this paper selects the water content as optimum water content -3%, optimum

Table 2. Basic parameters of phosphogypsum.

| Specific surface area / m$^2$·kg$^{-1}$ | Loss on ignition /% | Moisture content /% | Alkali content /% | Density /g·cm$^{-3}$ | fineness /% |
|---|---|---|---|---|---|
| 102 | 18.43 | 5.3 | 1.31 | 2.38 | 44.3 |

**Table 3. Chemical composition of phosphogypsum.**

| Ingredient | $SO_3$ | CaO | $SiO_2$ | $P_2O_5$ | $Na_2O$ | $Al_2O_3$ | Other |
|---|---|---|---|---|---|---|---|
| Mass fraction/% | 49.070 | 40.070 | 5.780 | 1.350 | 0.587 | 0.435 | 2.708 |

water content and optimum water content +3%. Dong Wei et al [25] found that after 6 times of wet-dry cycles, the red clay will produce obvious cracking, and the compressive strength of the soil body of the roadbed usually tends to stabilize after 4–6 times of wet-dry cycles [26], this test is based on the increase in the number of wet-dry cycles, so this test is formulated to be 7 times of wet-dry cycles.

When making samples, first dry the red clay soil samples and phosphogypsum, after drying, respectively, through the 2mm sieve, using alcohol combustion method to determine its moisture content after drying in order to reduce the sampling moisture content error. Weigh a certain mass of soil samples and phosphogypsum, calculate the amount of water required to formulate the mixture to the target moisture content. Weigh the required amount of water and add it to the mixture and mix it evenly, smother the material for 24h and then make samples according to the static compaction method in the "Highway Geotechnical Test Procedure" (JTG 3430–2020), the details of sample making are shown in Table 7.

### 3.2 Dry-wet cycle test method

Li Zhen, Xie Huihui et al [27,28] found that the wet-dry cycle path has a greater effect on the strength of red clay, and the degree of attenuation of the unconfined compressive strength of red clay by wetting and then drying is much greater than that of drying and then wetting, therefore, the wet-dry path of this test was chosen as wetting and then drying first. Referring to the research results in the literature [29–32]: after the roadbed is compacted with the optimum moisture content, the moisture content of the roadbed soil will be increased after a period of time in the natural climatic environment and fluctuates periodically or non-periodically within the range of 5% above and below the "equilibrium moisture content" (EMC). This paper further increases the wet-dry range on this basis, and the moisture content is +7% of the optimal moisture content when humidified, and -7% of the optimal moisture content when dry. The dry-wet cycle process is shown in Fig 1. The well-mixed mixture was made into a ring knife specimen, after the preparation was completed, the specimen was allowed to be maintained in an environment with a humidity of not less than 95% and a temperature of 20±2˚C for seven

**Table 4. Test results of heavy metals and radioactivity in phosphogypsum.**

| Test items | | Standard limits | Result | Conclusion |
|---|---|---|---|---|
| Heavy metal | Cu/mg·L$^{-1}$ | ≤100 | 0.157 | Qualified |
| | Zn/mg·L$^{-1}$ | ≤100 | 0.051 | Qualified |
| | Cd/mg·L$^{-1}$ | ≤1 | 0 | Qualified |
| | Pb/mg·L$^{-1}$ | ≤5 | 0 | Qualified |
| | Cr/mg·L$^{-1}$ | ≤15 | 0 | Qualified |
| | As/mg·L$^{-1}$ | ≤5 | 0.0356 | Qualified |
| | Hg/mg·L$^{-1}$ | ≤0.1 | 0.0005 | Qualified |
| Radioactivity | Ra-226/Bq·kg$^{-1}$ | — | 53.94 | — |
| | TH-232/Bq·kg$^{-1}$ | — | 42.13 | — |
| | K-40/Bq·kg$^{-1}$ | — | 52.95 | — |
| | $I_{Ra}$ | ≤1.0 | 0.3 | Qualified |
| | $I_\gamma$ | ≤1.0 | 0.3 | Qualified |

**Table 5. Basic parameters of cement.**

| Item | Index | Item | Index | Item | Index |
|---|---|---|---|---|---|
| 3d $f_{cf}$/MPa | 5.0 | Loss on ignition /% | 1.58 | Initial setting time /min | 302 |
| 28d $f_{cf}$/MPa | 6.7 | Alkali /% | 2.42 | Final setting time /min | 322 |
| 3d $f_{cu}$/MPa | 24.9 | Chloride ion /% | 0.018 | Stability | Qualified |
| 28d $f_{cu}$/MPa | 43.7 | Sulfur trioxide /% | 2.87 | | |

days, and after the maintenance was completed, the dry-wet cycle test was carried out in accordance with the method in Fig 2. The expansion and contraction test under dry-wet cycle is directly carried out in the soil expansion meter. Expansion and contraction test under dry and wet cycle is divided into two stages: (1) Humidification stage: humidify the prepared specimen to the target water content, calculate the corresponding amount of water, inject water with a syringe and leave it for 24h so that the water content inside the soil sample is uniform, and record the readings after completion. (2) Drying stage: After humidification, turn on the heater until the soil sample is dried to the target water content, leave it for 24h, and record the reading after drying.

### 3.3 Expansion and contraction test

1 wet-dry cycle includes humidification expansion deformation and drying shrinkage deformation. For the humidification expansion process, the absolute expansion rate $\delta_{ae}$ is used to characterize the process, and for the drying shrinkage process, the absolute shrinkage rate $\delta_{as}$ is used to characterize the process. Eqs (1) and (2) are used to calculate, respectively.

$$\delta_{ae} = \frac{h_{ei} - h_0}{h_0} \times 100\% \tag{1}$$

$$\delta_{as} = \frac{h_{si} - h_0}{h_0} \times 100\% \tag{2}$$

Note: $h_{ei}$ = height of the specimen after the ith expansion stabilisation; $h_0$ = initial height of the specimen; $h_{si}$ = height of the specimen after the i-th contraction stabilisation.

### 3.4 Fracture test

The fissure test can be synchronized with the expansion and contraction test, and at the end of each dry and wet cycle, photographs are taken to record the fissure development of the

**Table 6. Optimal moisture content and maximum dry density of mixture under different proportion and cement content.**

| P:T | 4% C | | 6% C | | 8% C | |
|---|---|---|---|---|---|---|
| | OMC/% | MDD/g·cm$^{-3}$ | OMC/% | MDD/g·cm$^{-3}$ | OMC/% | MDD/g·cm$^{-3}$ |
| 1:1 | 21.3 | 1.559 | 21.8 | 1.555 | 20.8 | 1.569 |
| 1:2 | 22.5 | 1.535 | 23.1 | 1.549 | 21.8 | 1.556 |
| 1:3 | 23.46 | 1.519 | 24.1 | 1.538 | 22.9 | 1.543 |
| 1:4 | 24.36 | 1.451 | 25.6 | 1.511 | 24.5 | 1.499 |
| 1:5 | 26.0 | 1.459 | 26.7 | 1.495 | 25.7 | 1.504 |

Note: In this paper, C stands for cement, P stands for phosphogypsum, T stands for red clay, OMC stands for optimum moisture content, MDD stands for maximum dry density, and PCAS stands for image processing system.

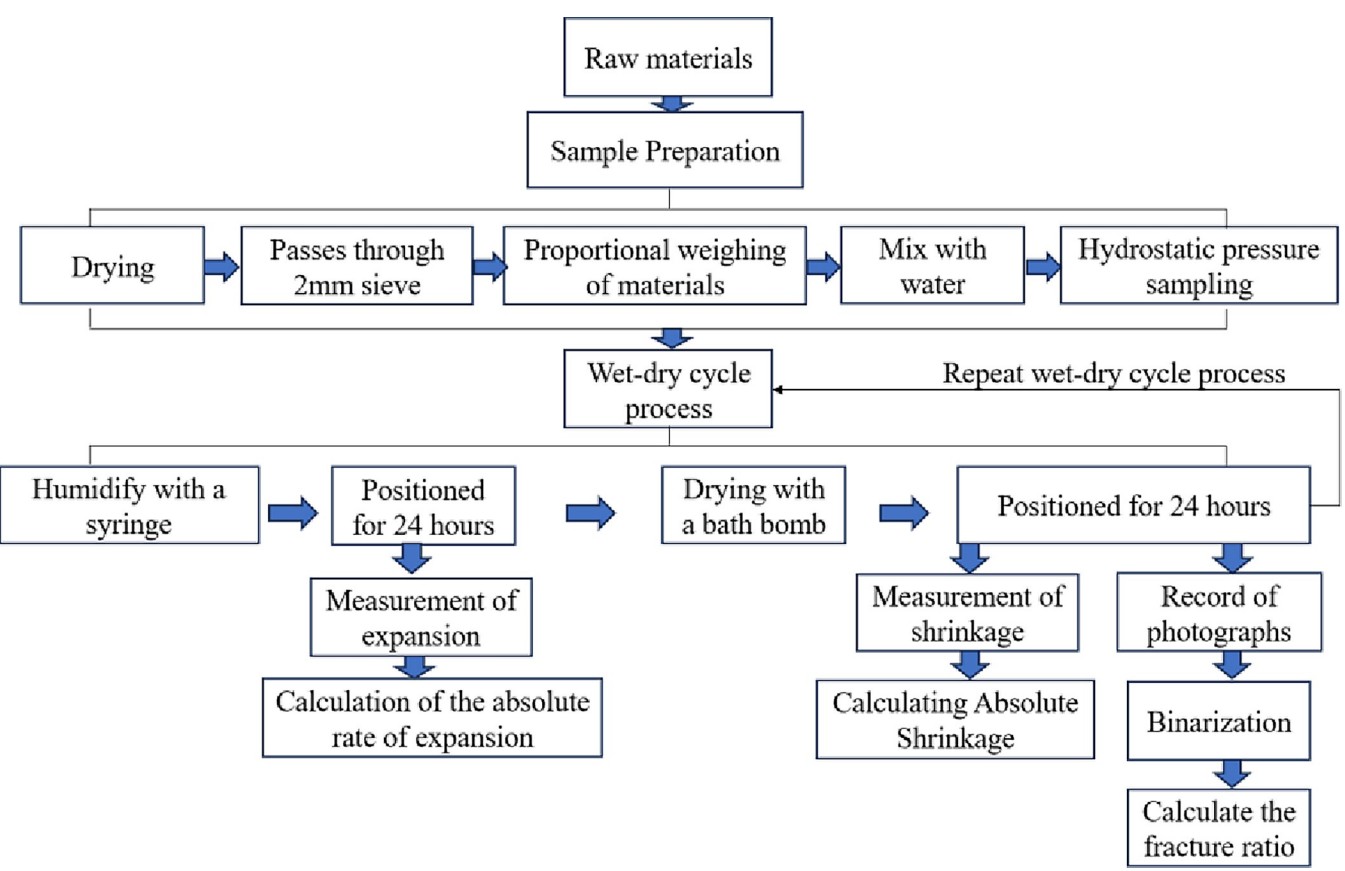

**Fig 1. Flowchart of the testing process.**

specimen, and the photographed pictures are processed with an image analysis system (PCAS), and the processing is shown in Fig 3: Firstly, the part of the captured fissure picture other than the specimen was cropped out, leaving the circular ring knife specimen surface; then the remaining picture was converted into a binarized gray and white image, adjusted its grayscale to make the fissure clear, and then removed the miscellaneous dots in the binarized image that were not related to the fissure, and finally the processed picture was described by the black and white pixel dots, where the black pixel dots served as the quantitative parameter for the fissure, and the sum of the black pixel dots characterized the fissure area, the total pixel points represent the total surface area of the specimen, and the fissure rate is the ratio of the fissure area to the total surface area of the specimen, which is calculated according to Eq (3).

$$\delta_{fA} = \sum_{i=1}^{i=n} A_i / A \tag{3}$$

**Table 7. Specimen preparation.**

| Batch number | Compaction /% | C/% | Moisture content /% | P:T | Number of wet-dry cycles /times | Number of samples /pcs |
|---|---|---|---|---|---|---|
| 1 | 90,92,93,94,95,96 | 6 | OMC | 1:1,1:2,1:31:4,1:5 | 1~7 | 30 |
| 2 | 93 | 4,6,8 | OMC | 1:1,1:2,1:3,1:4,1:5 | 1~7 | 15 |
| 3 | 93 | 6 | OMC-3%,OMC, OMC +3% | 1:1,1:2,1:3,1:4,1:5 | 1~7 | 15 |

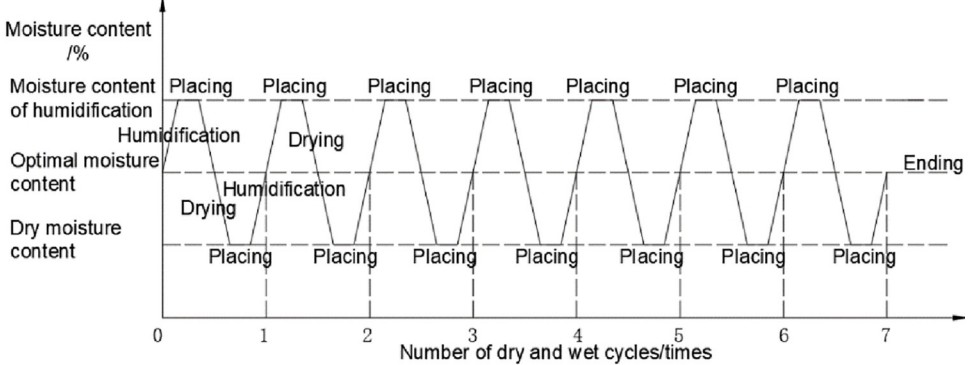

**Fig 2. Schematic diagram of the wet-dry cycle process.**

Note: $\delta_{fA}$ = fissure rate/%; A = sample area/$mm^2$; $A_i$ = area occupied by the i-th fissure/$mm^2$.

## 4 Analysis of test results

### 4.1 Absolute expansion rate of the mix

**4.1.1 Relationship between absolute expansion rate and degree of compaction.** The variation of absolute expansion of specimens with compaction degree at 6% cement dosing and optimum water content is shown in Fig 4. From Fig 4, it can be seen that with the increase of compaction degree, the absolute expansion rate of the specimen is generally into an increasing trend, with P: T = 1:4, seven wet-dry cycles, for example, when the compaction degree of the specimen is increased from 90% to 96%, its absolute expansion rate is increased from 10.8% to 17.2%, an increase of 59.3%. Lv Haibo et al [33] concluded that the top three mineral components in red clay were kaolinite ($Al_2Si_2O_5(OH)_4$), clinopyroxene (FeO(OH)), and alumina trihydrate ($Al(OH)_3 \cdot 3H_2O$) through the physical phase analysis of red clay, in which $Al(OH)_3 \cdot 3H_2O$ is hydrophilic, and at the same time, phosphogypsum contained the hydrophilic mineral $Na_2SiF_6$, in the preparation of the specimen, the greater the compaction of the specimen, the smaller the spacing between the soil particles, containing more soil particles in the same volume, the hydrophilic mineral content contained in the mixed material increases, enhancing the expansion potential within the specimen, so that the compaction of the specimen increases the amount of expansion when the specimen absorbs water to increase the humidity. On the contrary, when the compaction of the specimen is small, the internal soil particles and hydrophilic materials are less, and the specimen swells less when it absorbs water and humidifies.

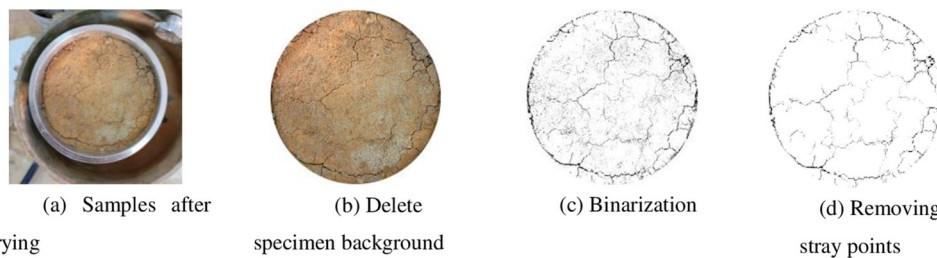

(a) Samples after drying
(b) Delete specimen background
(c) Binarization
(d) Removing stray points

**Fig 3. Image processing process.**

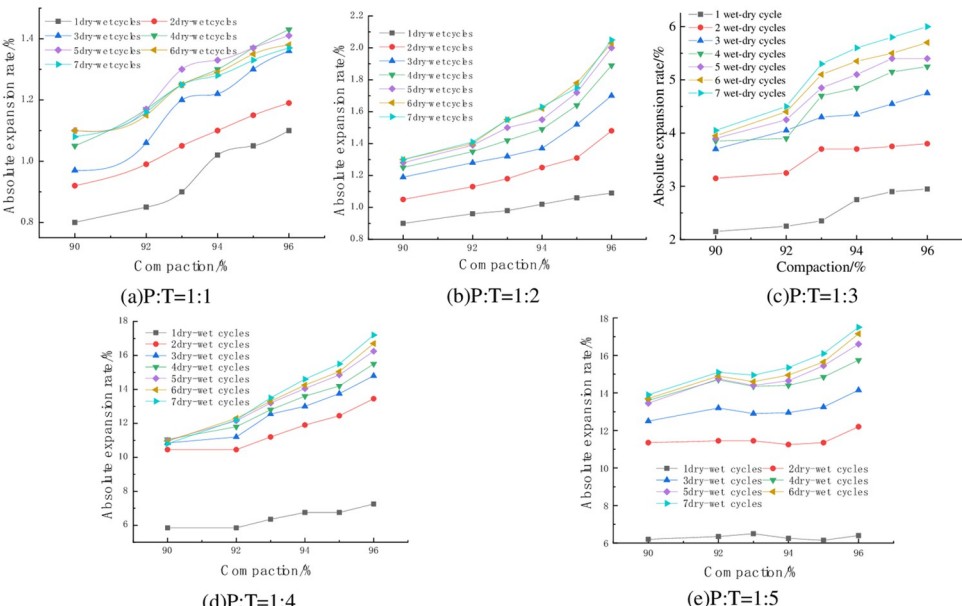

**Fig 4. Relationship between absolute expansion and compaction.**

**4.1.2 Relationship between absolute expansion rate and number of wet-dry cycles.** The absolute expansion of the specimens with 6% cement dosing, optimum water content, and different numbers of wet-dry cycles is shown in Fig 5. As can be seen from Fig 5, the absolute expansion rate of the specimen with the increase of the number of wet-dry cycles showed three stages: rapid growth—slow growth—stabilization. The expansion deformation is mainly concentrated in the first three dry-wet cycles, and stabilized after 6~7 times. This is because in the process of 1 ~ 3 dry-wet cycles, the closed pores of the soil sample gradually transform into connected pores, the porosity increases rapidly, and the microscopic stress-strain field changes, and cracks are generated inside the soil sample [34,35], resulting in an increase in the absolute expansion rate on the macro level. However, as the number of dry-wet cycles increases to 6 ~ 7 times, the porosity and crack size and number change little. Therefore, the absolute expansion rate of the mixture tends to be stable after 6 ~ 7 dry-wet cycles.

**4.1.3 Relationship between absolute expansion rate and cement dosage.** The absolute expansion of phosphogypsum stabilized red clay specimens at 93% compaction, optimum moisture content, and different cement dosages is shown in Fig 6 From Fig 6, it can be seen that the absolute expansion rate of the mix is positively correlated with the dosage of cement. Taking P:T = 1:4 and three dry—wet cycles as an example, the absolute expansion rate is 9.95% when the cement dosage is 4%, and 12.55% when the cement dosage is 6%, which is an increase of 26.13% in the absolute expansion rate. This is because when the CaO in cement and the $SiO_2$ in clay meet with water, hydration reaction occurs to generate hydrated calcium silicate (C-S-H) gel, hydrated calcium aluminate (C-A-H) and $Ca(OH)_2$ [36], respectively. It has a certain hydrophilicity. The ion exchange of reactants will promote the aggregation of particles inside the mixture and fill the pores between particles. In the process of water absorption and humidification of the sample, the adsorption capacity of the mixture to water molecules is enhanced, thereby increasing the water absorption expansion of the mixture.

**4.1.4 Relationship between absolute expansion rate and P:T.** The changes of absolute expansion rate of specimens under optimum water content, 93% compaction, and different P: T are shown in Fig 7. It can be seen from Fig 7 that when the compactness of the sample is

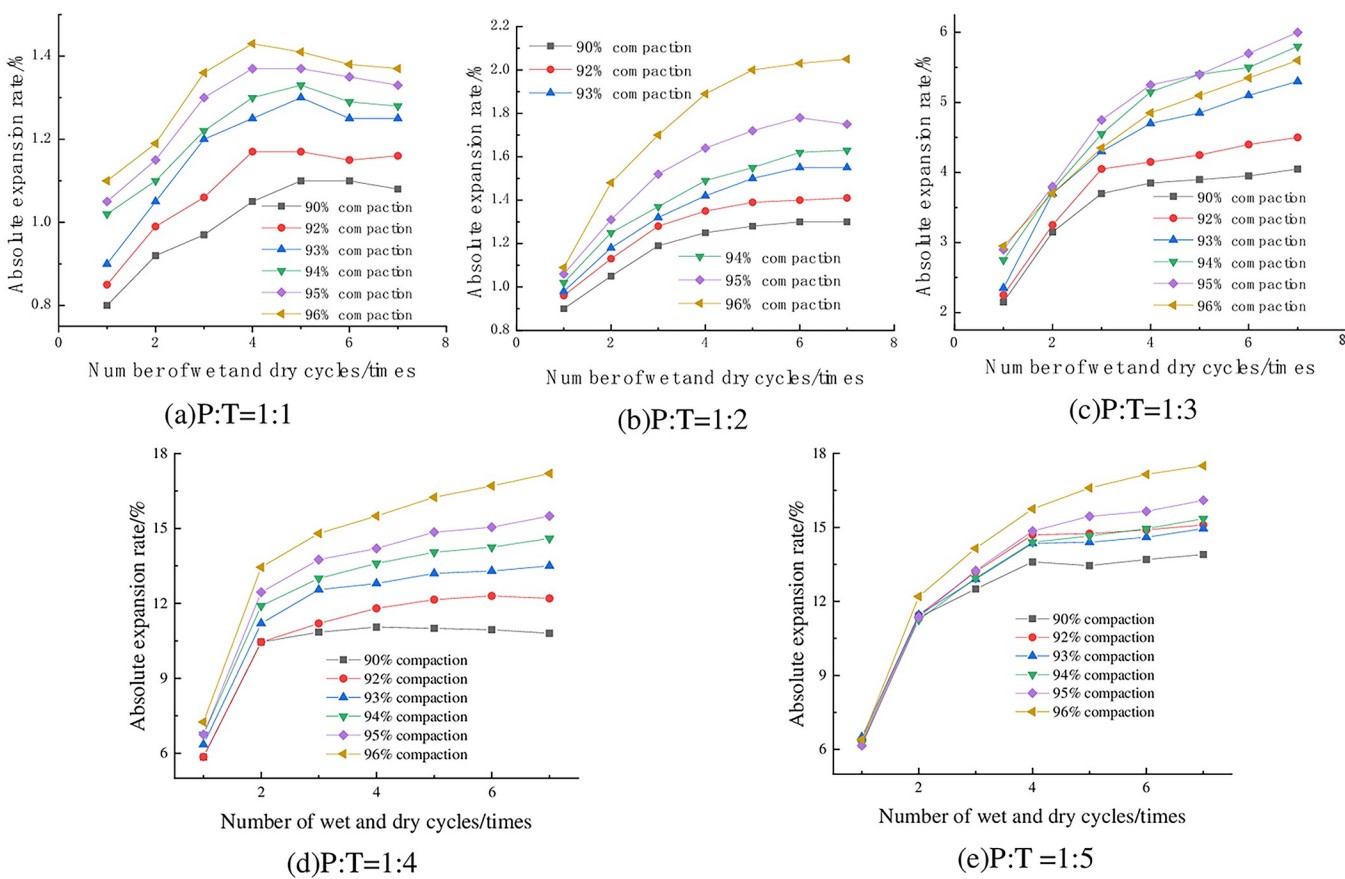

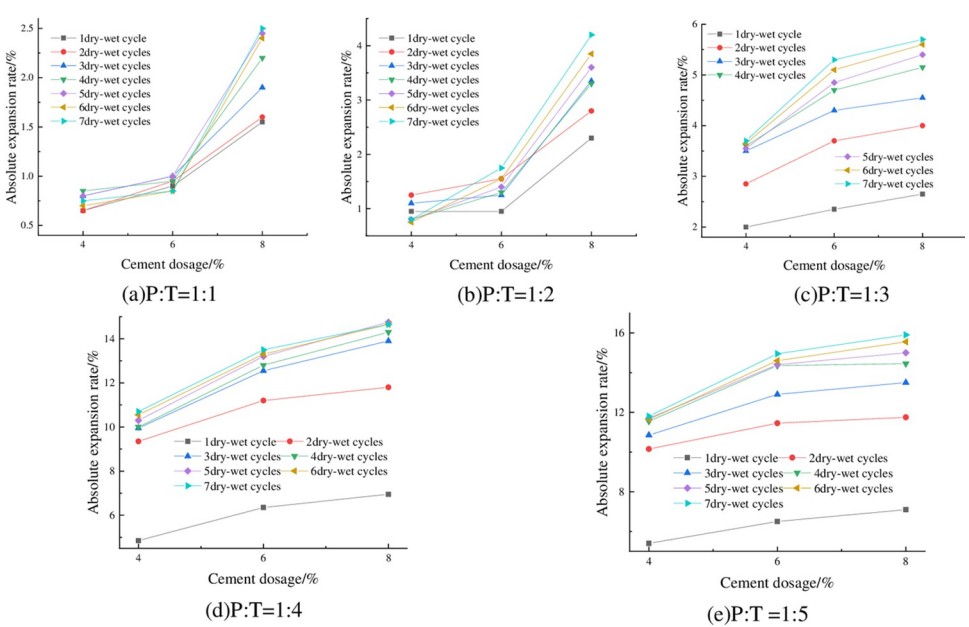

**Fig 5. Absolute expansion rate versus number of wet-dry cycles.**

**Fig 6. Relationship between absolute expansion rate and cement dosage.**

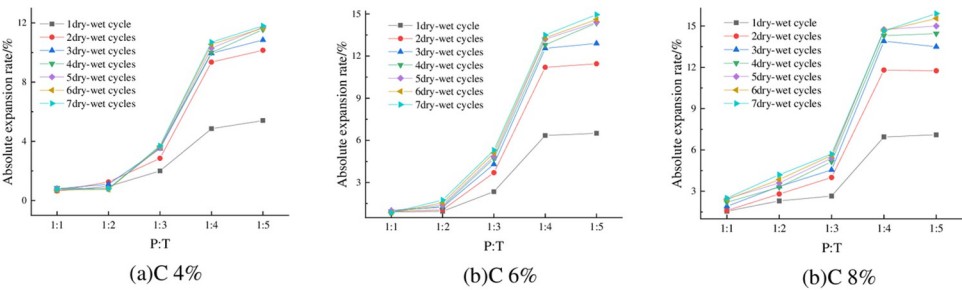

**Fig 7. Relationship between absolute expansion rate and P: T.**

constant, the absolute expansion rate is negatively correlated with P: T, that is, the higher the content of phosphogypsum, the smaller the absolute expansion rate of the sample. Taking 6% cement dosage, 93% compaction degree, 5 times of wet-dry cycles as an example, when P: T decreased from 1:3 to 1:4, its absolute expansion rate also increased from 4.85% to 13.2%, with an increase of 172.16%, which is more prominent. From the above law, it can be seen that phosphogypsum has obvious inhibition effect on the expansion and deformation of red clay, and the more phosphogypsum doping, the better the inhibition effect, the mechanism is as follows: (1) Cement hydration generates a large number of layered hydrated calcium silicate gel (C-S-H), the reaction process is shown in the Eq (4), $CaSO_4·2H_2O$ in phosphogypsum reacts with the cement hydration product of $Ca(OH)_2$, $3CaO·Al_2O_3$ to generate needle-like calcium alumina (AFt), the reaction process is shown in the Eqs (4) and (6), and the product fills in the internal pores of the mix, which improves the structure of the red clay and thus reduces the space for its water absorption and expansion; (2) phosphogypsum fills in between the particles of the red clay, which improves the densification of the red clay, and thus inhibits its expansion and deformation [37]; (3) the main mineral components of the red clay are kaolinite (3) The main mineral components of red clay are kaolinite, montmorillonite and illite, of which montmorillonite has strong hydrophilicity, and it is easy to swell with water absorption and shrink with water loss, while phosphogypsum is mainly composed of calcium sulfate dihydrate, and calcium sulfate dihydrate has poorer hydrophilicity, therefore, with the increase of the phosphogypsum content, the montmorillonite content of the mix decreases, and the content of calcium sulfate dihydrate increases, which results in a decrease of the swelling and deformation.

$$3CaO \bullet SiO_2 + n\ H_2O- \rightarrow x\ CaO \bullet S_iO_2 \bullet (n-3+x)H_2O + (3-x)Ca(OH)_2 \qquad (4)$$

$$3Ca(OH)_2 + Al_2O_3 + (n-3)H_2O- \rightarrow 3CaAl2O3 \bullet n\ H_2O \qquad (5)$$

$$3CaO \bullet Al_2O_3 + 3(CaSO_4 \bullet 2H_2O) + 2Ca(OH)_2 + 24H_2O- \rightarrow 3CaO \bullet Al_2O_3 \bullet 3CaSO_4 \bullet 32H_2O \ (6)$$

**4.1.5 Relationship between absolute expansion rate and initial moisture content.**   The absolute expansion rate of the samples with P: T = 1: 1, P: T = 1: 3 and P: T = 1: 5 under 6% cement content, 93% compaction degree and different initial water content is shown in Fig 8. From Fig 8, it can be seen that the absolute expansion rate of the specimens decreases with the increase of the initial moisture content and is close to a linear relationship. Taking P: T = 1: 3 and 7 dry-wet cycles as an example, when the moisture content of the sample increases from 21% to 24%, the absolute expansion rate changes from 7.85% to 5.3%, which is reduced by 32.48%. The reason is that when the sample is prepared, the larger the initial moisture content is, the particles of the mixture are fully water-absorbed and expanded after 24 h, and the

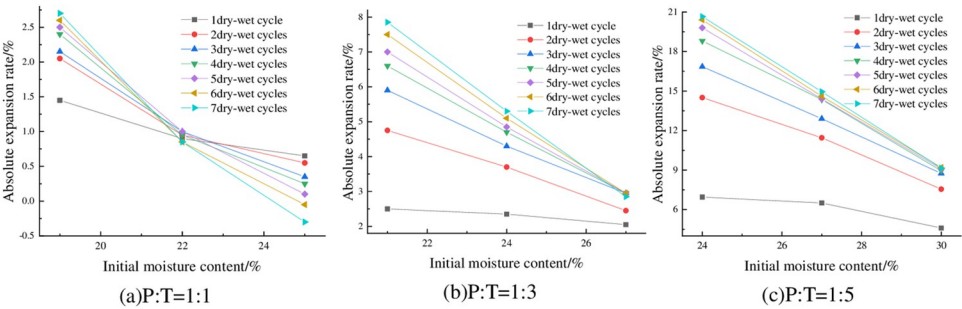

**Fig 8. Relationship between absolute expansion and initial moisture content.**

expansion potential is released in advance. The residual expansion potential is small, so the absolute expansion rate of the sample decreases after increasing the initial moisture content.

## 4.2 Absolute shrinkage rate of the mix

### 4.2.1 Relationship between absolute shrinkage rate and degree of compaction.

The absolute shrinkage of the specimens at optimum moisture content, 6% cement dosing and different compaction degrees is shown in Fig 9. From Fig 9, it can be seen that the absolute shrinkage increases with compaction degree. Taking P: T = 1: 2 and 4 dry-wet cycles as an example, when the compaction degree of the sample increases from 90% to 92%, the absolute shrinkage rate changes from-0.24% to 0.25%, and the absolute expansion rate changes from negative to positive, which indicates that the sample after drying has a slight expansion deformation. From Fig 9(C)-9(E), it can be seen that the absolute shrinkage of the specimens is greater than zero, indicating that when P: T is less than 1:3, with the increase of compaction, the specimen not only does not produce shrinkage deformation, but also the expansion deformation is getting bigger and bigger.

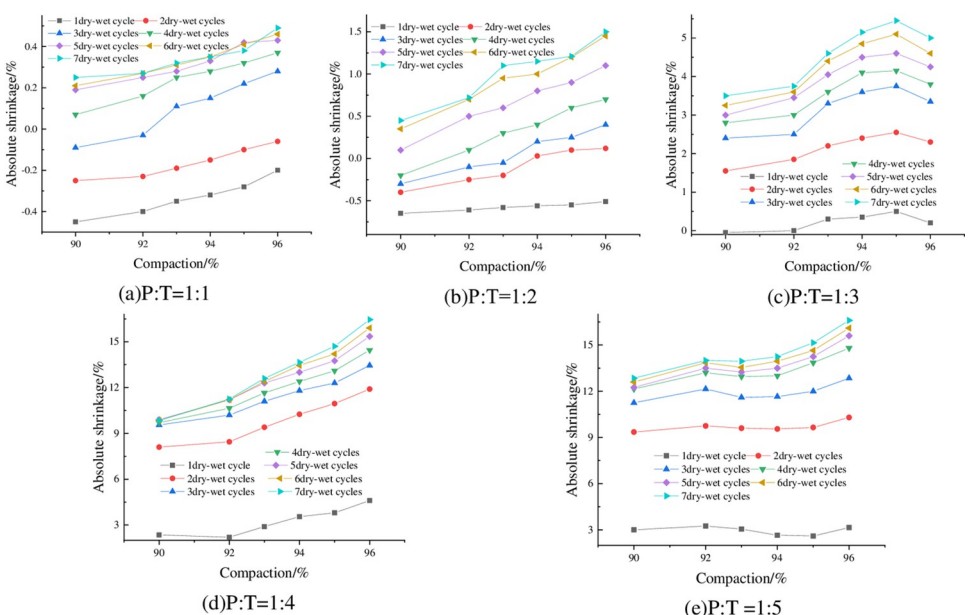

**Fig 9. Relationship between absolute shrinkage rate and compaction.**

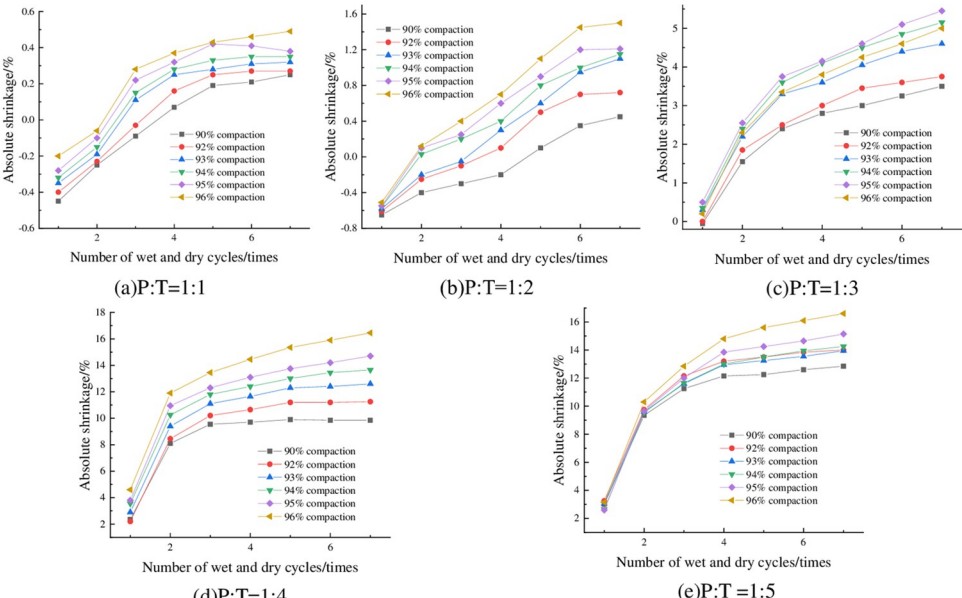

**Fig 10. Relationship between absolute shrinkage and number of wet-dry cycles.**

**4.2.2 Relationship between absolute shrinkage rate and the number of wet-dry cycles.** The absolute shrinkage of the specimens with optimum moisture content, 6% cement dosing and different number of wet-dry cycles is shown in Fig 10. From Fig 10, it can be seen that the absolute shrinkage of the specimens increased with the increase of the number of wet-dry cycles, and all of them showed a tendency of rapid increase first, and then gradually leveled off. From Fig 10(A) and 10(B), it can be seen that with the increase of the number of wet-dry cycles, the absolute shrinkage of the specimen is changed from negative to positive, and then stabilized, which indicates that with the increase of the number of wet-dry cycles, the shrinkage deformation of the specimen is getting smaller and smaller, and continue to increase the number of wet-dry cycles, then the expansion deformation is generated, and expansion deformation with the increase of the number of wet and dry cycles is increased gradually and then tends to be stabilized.

**4.2.3 Relationship between absolute shrinkage rate and cement dosage.** The absolute shrinkage of the specimens at 93% compaction, optimum moisture content and different cement dosage is shown in Fig 11. As can be seen from Fig 11, the absolute shrinkage of the specimens increased with the increase of cement dosing. Taking P:T = 1:4 with 3 wet-dry cycles as an example, when the cement dosage was increased from 4% to 6%, the absolute shrinkage increased from 8.6% to 11.1%, which is an increase of 22.52%.

**4.2.4 Relationship between absolute shrinkage rate and P: T.** The absolute shrinkage of the specimens under 93% compaction, optimum moisture content, and different P: T is shown in Fig 12. From Fig 12, it can be seen that the absolute shrinkage of the specimen is negatively correlated with P:T, that is, with the increase of phosphogypsum content, the absolute shrinkage decreases, and the absolute shrinkage of the specimen is basically greater than zero, which indicates that the specimen almost does not occur under the action of the wet-dry cycles of vertical shrinkage deformation, but vertical expansion deformation, and the higher the phosphogypsum mixing, the better the deformation inhibition effect.

**4.2.5 Relationship between absolute shrinkage rate and initial moisture content.** The absolute shrinkage of the specimens under 93% compaction, 6% cement dosing and different

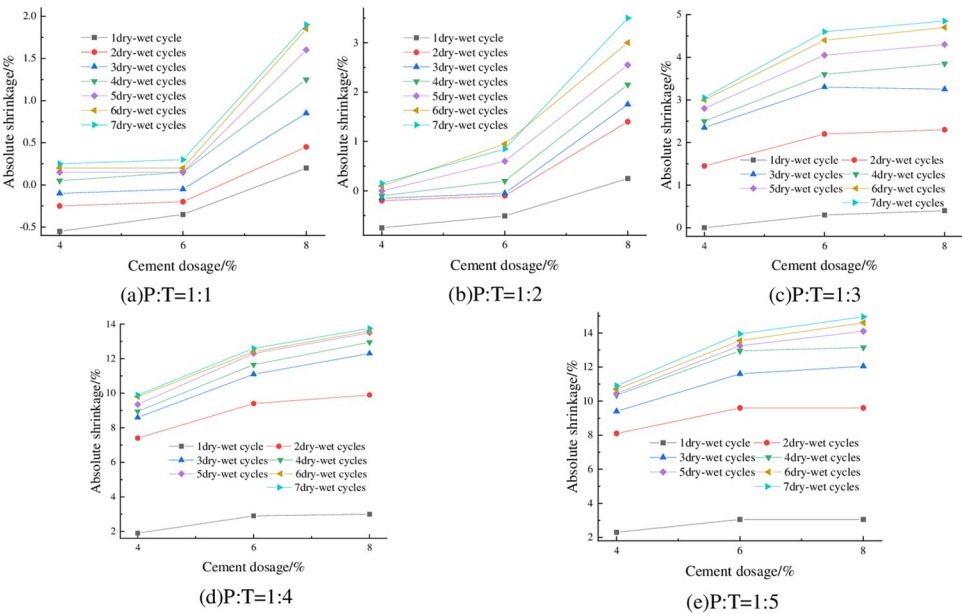

**Fig 11. Relationship between absolute shrinkage and cement dosage.**

initial water contents is shown in Fig 13. As can be seen from Fig 13, when P: T is certain, its absolute expansion rate is negatively correlated with the initial moisture content, and the absolute shrinkage rate is almost always greater than zero under different numbers of wet-dry cycles, which also indicates that the specimen almost does not undergo vertical shrinkage deformation but produces vertical expansion deformation in the process of wet-dry cycles, which is in line with the previous law, indicating that the height of the specimen is higher than the initial height at the end of each humidification and drying. Taking P: T = 1:3 and 4 wet-dry cycles as an example, when the initial moisture content of the specimen was increased from 21% to 24%, its absolute shrinkage also decreased from 5.55% to 3.6%, which is a decrease of 35.14%.

## 4.3 Fracture extension pattern

Taking 96% compactness, 6% cement content, P: T = 1: 1 and P: T = 1: 5 as examples, the generation and expansion rules of cracks in phosphogypsum stabilized red clay materials under dry-wet cycles are qualitatively compared and analyzed, as shown in Figs 14 and 15. It can be seen from Fig 14 that after the first dry-wet cycle, the sample did not produce obvious cracking

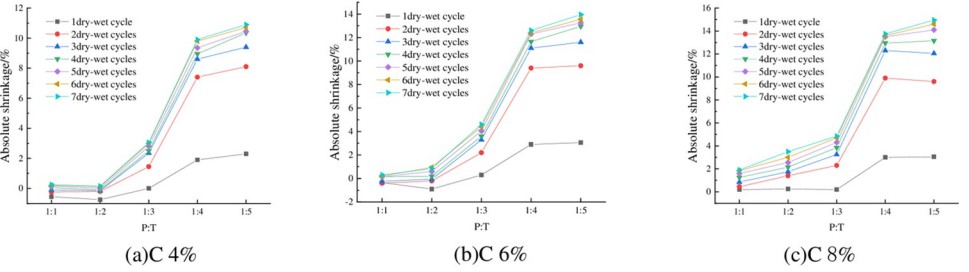

**Fig 12. Relationship between absolute shrinkage and P: T.**

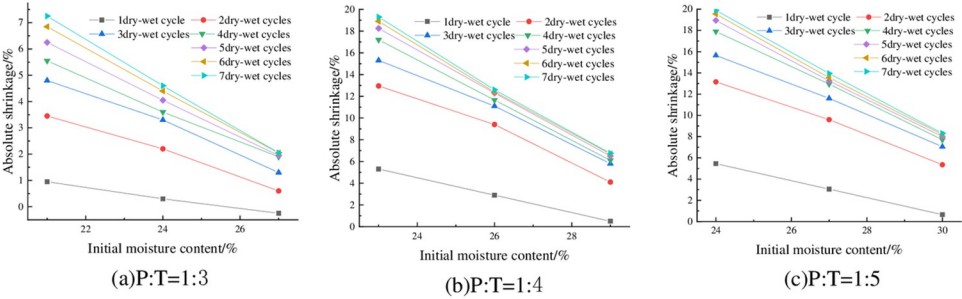

**Fig 13. Relationship between absolute shrinkage rate and initial moisture content.**

deformation. After two dry-wet cycles, the sample began to crack slightly, and the cracks were mainly produced at the edge of the sample, and the edge was accompanied by slight soil block shedding. After 3 ~ 4 times of wet-dry cycles, the cracks at the edge of the sample were widened, deepened and extended on the basis of the previous two times, and slight cracks began to appear near the center of the sample. After 5 and 6 dry-wet cycles, the cracks of the sample gradually extended to the center of the sample, and the width and length of the cracks further increased. After 7 dry-wet cycles, the width of the cracks at the edge of the sample remained basically unchanged. This is due to the restraint effect of the ring cutter on the surroundings of the sample during the test, resulting in no further increase in the width of the cracks, while the cracks in the central area increased compared with before. From Fig 15, it can be seen that when P: T = 1: 1, there are no obvious cracks on the surface of the sample after 7 dry-wet cycles, only a small amount of soil blocks fall off at the edge. With the increase of the number of dry-wet cycles, the degree of shedding is getting higher and higher. By comparing the crack propagation process of the two, it can be seen that the higher phosphogypsum content can effectively inhibit the crack development of red clay.

The fracture rate of the specimens under 6% cement dosing, optimum water content, and different numbers of wet-dry cycles are shown in Fig 16. Limited to space, only the results of P: T = 1: 4 and P: T = 1:5 is shown here, and other ratios have similar laws with them, which will not be repeated in this paper. As can be seen from Fig 16, the fracture rate of the specimen increases with the increase of the number of wet-dry cycles, and then gradually tends to level off. This is consistent with the absolute shrinkage of the specimen with the increase in the number of wet-dry cycles, showing the first rapid increase, and then gradually tend to flatten the rule of change (Fig 10). The reason is that the specimen produces tensile stress during the contraction process, and when the tensile stress is greater than the tensile strength of the specimen, cracks will be produced. With the increase of the number of wet-dry cycles, the particles become more and more compact with each other, the internal structure is stabilized, and the cleavage rate tends to stabilize. Taking P: T = 1:5 and 96% compaction as an example, when

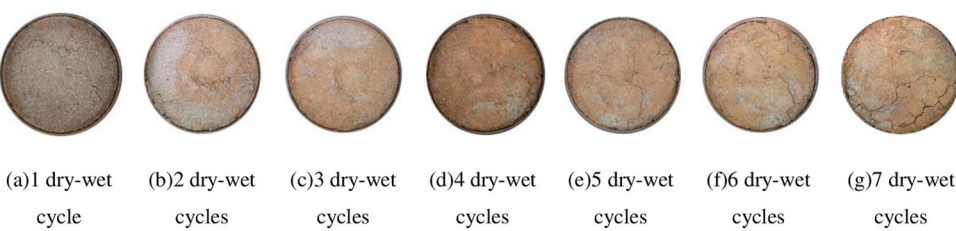

| (a)1 dry-wet cycle | (b)2 dry-wet cycles | (c)3 dry-wet cycles | (d)4 dry-wet cycles | (e)5 dry-wet cycles | (f)6 dry-wet cycles | (g)7 dry-wet cycles |

**Fig 14. P: T = 1:5 Fracture development in 7 wet-dry cycles.**

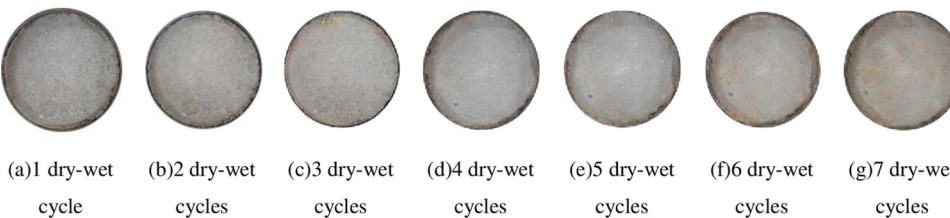

| (a)1 dry-wet cycle | (b)2 dry-wet cycles | (c)3 dry-wet cycles | (d)4 dry-wet cycles | (e)5 dry-wet cycles | (f)6 dry-wet cycles | (g)7 dry-wet cycles |

**Fig 15. P: T = 1: 1 Fracture development in 7 wet-dry cycles.**

the number of wet-dry cycles increased from 2 to 3, the cleavage rate changed from 1.89% to 2.66%, an increase of 40.74%.

The fracture rate of the specimens with 6% cement admixture, optimum water content and different compaction degree is shown in Fig 17. As can be seen from the figure, the fracture rate of the specimens is getting smaller and smaller with the increase of compaction degree. Taking 4 dry-wet cycles and P: T = 1:5 as an example, when the compaction degree of the specimen is increased from 92% to 96%, the fracture rate decreases from 4.91% to 3.56%, which is a decrease of 27.49%.

The fracture rates of the specimens at optimum water content, 6% cement dosing and different P: T are shown in Fig 18. As can be seen from Fig 18, the fracture rate of the specimens gradually increased with the decrease of P: T, that is, with the decrease of phosphogypsum dosage, the fracture rate showed an increasing trend. It can be seen that phosphogypsum has a significant inhibitory effect on the cracking and deformation of red clay. Taking 90% compaction and 4 wet-dry cycles as an example, when P: T changed from 1:4 to 1:5, the fracture rate of the specimen increased from 4.69% to 5.32%, which was an increase of 13.43%.

The fracture rate of the specimens at 93% compaction, optimum moisture content and different cement dosage is shown in Fig 19. From Fig 19 it can be seen that the fissure ratio decreases with the increase of cement dosage. This is because when the cement dosing is higher, the cement, phosphogypsum and red clay react to generate more hydrated calcium silicate and calcium alumina and other substances with strong adsorption capacity, cementing a large number of soil particles, the cement intertwined with each other, the large volume of pores is converted to small pores, and the degree of densification of the material is increased, forming a stable, dense overall skeleton structure, thus improving the tensile strength of the specimen; at the same time, calcium alumina has a micro-expansion, to a certain extent, can

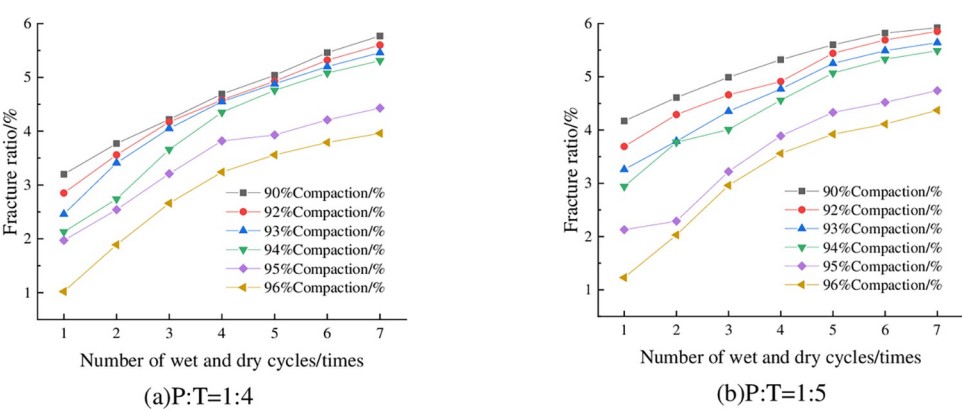

(a)P:T=1:4                    (b)P:T=1:5

**Fig 16. Relationship between fracture ratio and number of dry-wet cycles.**

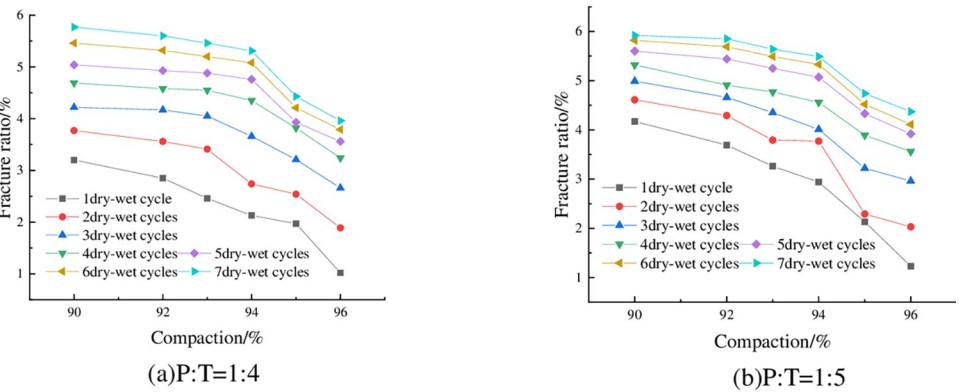

**Fig 17. Relationship between fracture ratio and compaction degree.**

fill the fissures of the mixed materials [38], so the increase of cement doping can reduce the fissure rate of the specimen. Taking P: T = 1:5 and 5 wet-dry cycles as an example, when the cement doping of the specimen was increased from 4% to 8%, the fracture rate decreased from 5.68% to 4.98%, which is a decrease of 12.32%.

The fracture rates of the specimens at 93% compaction, 6% cement admixture and different initial water contents are shown in Fig 20. As can be seen from Fig 20, the fracture rate showed a slight decreasing trend with the increase of initial water content. Taking P: T = 1:5 and 6 wet-dry cycles as an example, the fracture rate was 5.47% when the initial water content was 24%, and 5.25% when the initial water content was 27%, with a decrease of 4.1%. This is because after the increase of water content, some of the fine particles formed inside the specimen adsorb the water film to wrap the microporous space, and then form a monolithic lamellar structure [39], which to a certain extent in advance of the release of the expansion potential, so that in the process of drying and contraction of the tensile stress generated by the smaller, at the same time, its tensile strength has been improved, and so the cracking rate of the specimen under the higher initial water content is smaller.

## 4.4 Fitting of absolute expansion (shrinkage) rate to compaction and fracture rate

The influencing factors and changing rules of expansion and contraction and cracking deformation of phosphogypsum stabilized red clay were derived from the previous paper, on the basis of which the mathematical relationship between deformation indexes and their

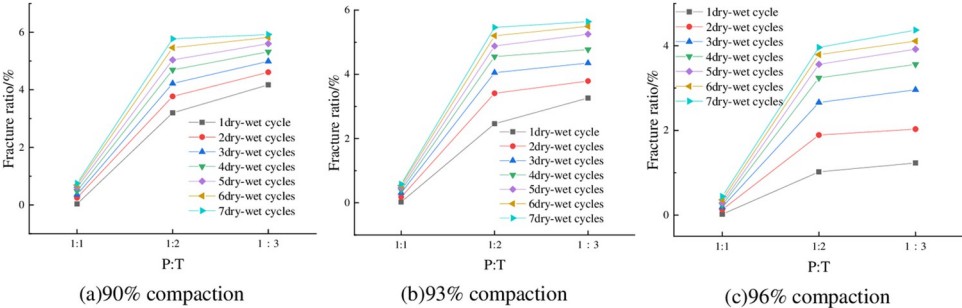

**Fig 18. Relationship between fracture rate and P: T.**

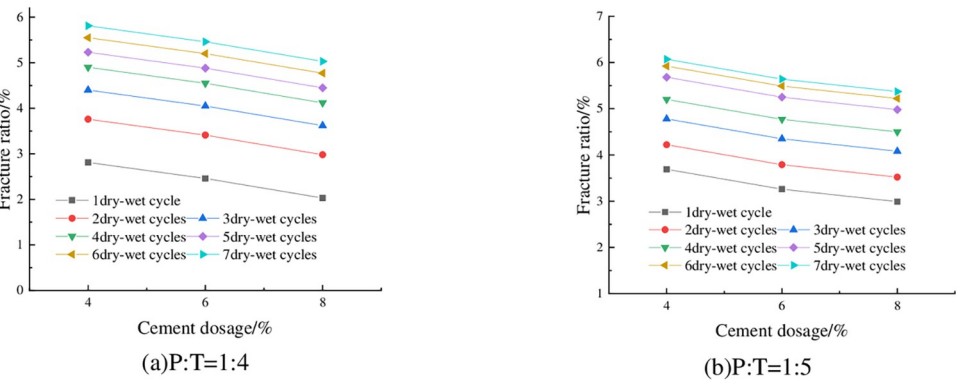

**Fig 19. Relationship between fracture ratio and cement content.**

influencing factors was further explored. Because compaction is an important parameter of roadbed filler, the mathematical relationship between compaction and the absolute expansion rate and cracking rate of the mixture is considered to be described by expressions. The software Origin2018 was applied to fit the data obtained from the test. Limited to space, only show the case of 6% cement mixing, optimal water content, P: T = 1:5, there are similar laws in other cases, which will not be repeated. The nonlinear fitting equations for compaction, absolute expansion and fracture rate are shown in Eq (7), and the absolute shrinkage of the mix also satisfies the law, and the fitting results are shown in Table 8 and Fig 21.

$$f(x, y) = ax + by + cx^2 + dy^2 + e \qquad (7)$$

Note: f(x,y) = fracture rate (%); $x$ = compaction (%); $y$ = absolute expansion (%); a,b,c,d, e = a,b,c,d,e are test parameters

In Table 8, $R^2$ is a fit index, and its value is closer to 1, which means the better the fit, and in general, its value is not less than 0.7, which means the fit is better. As can be seen from Table 8, the values of $R^2$ in the fit are all greater than 0.84, indicating that this nonlinear quadratic equation can fit the relationship between compaction, fracture rate and absolute expansion and contraction or shrinkage better.

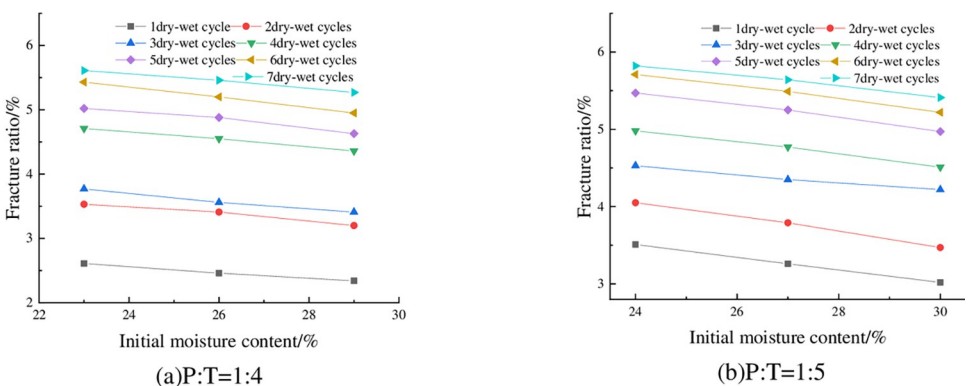

**Fig 20. Relationship between fracture ratio and initial water content.**

**Table 8. Fitting results.**

| Coefficient | $\delta_{ae}$ | $\delta_{as}$ |
|---|---|---|
| a | 15.01025 | 13.2323 |
| b | -0.4023 | -0.12707 |
| c | -0.08332 | -0.07336 |
| d | 0.02978 | 0.0184 |
| e | -670.59986 | -592.44555 |
| $R^2$ | 0.94516 | 0.84533 |

## 5 Conclusion

In this paper, the expansion and contraction performance and the crack expansion law of phosphogypsum stabilized red clay under the action of wet-dry cycles were investigated through the expansion and contraction experimental test and the image processing system PCAS, and the following conclusions were obtained:

1. The absolute expansion rate and absolute shrinkage rate of cement-phosphogypsum-red clay are positively correlated with the compaction degree, the number of dry and wet cycles and the cement dosage, and negatively correlated with the initial water content and phosphogypsum dosage;

2. Cement-phosphogypsum-red clay fracture rate increases with the increase of the number of dry and wet cycles, and decreases with the increase of the initial water content, compaction, cement and phosphogypsum dosage;

3. The relationship between absolute expansion or shrinkage of cement-phosphogypsum-red clay and compaction degree and fissure ratio can be fitted by a nonlinear quadratic equation $f(x,y) = ax+by+cx^2+dy^2+e$, with a,b,c,d,e as test constants.

4. Phosphogypsum has obvious inhibiting effect on the expansion, contraction and cracking of the mixture, and it is suggested that the cement doping is 6% and phosphogypsum:red clay = 1:1~1:2 as the roadbed filler.

The main ways of comprehensive utilization of phosphogypsum resources are mine filling and ecological restoration materials, road construction materials, soil conditioning agent, cement retarder, gypsum-based building materials, sulfuric acid, chemical fillers, etc., but the

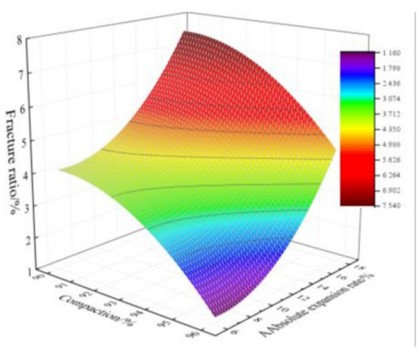

(a) Absolute expansion rate fitting          (b) Absolute shrinkage fitting

**Fig 21. Fitting results.**

utilization rate of phosphogypsum as a whole is at a low level. Phosphogypsum-processed road-building materials are used in highway water stabilization layer and other technically feasible, and there are successful test sections at home and abroad. Restricted by the temporary absence of successful experience in large-scale application, as well as environmental protection and other administrative licensing and related policy support and other factors, phosphogypsum as road construction materials has not yet opened the market, not to achieve engineering, large-scale application.

1. phosphogypsum in the road project application of low admixture, so that the increasing production of phosphogypsum still exists "oversupply" of the status quo, phosphogypsum accumulation problem can not be effectively solved, phosphogypsum utilization is low, an important reason for the lack of technological support capacity for the comprehensive utilization of phosphogypsum, basic, prospective research and development of technologies The investment is seriously insufficient, and there is no breakthrough in the comprehensive utilization technology of large dosage. There is no systematic standard system covering phosphogypsum emissions, comprehensive utilization technology, comprehensive utilization products and other aspects.

2. It is recommended to increase the study of the impact of phosphogypsum on the environment, the relevant departments to clarify the environmental protection and other relevant administrative licenses as a basis for relevant policies and safeguards to guide, support and support.

## Supporting information

**S1 Data set. Minimum data set.**
(DOCX)

## Author Contributions

**Conceptualization:** Kaisheng Chen.

**Funding acquisition:** Kaisheng Chen.

**Software:** Zeyu Liu.

**Supervision:** Zeyu Liu.

**Writing – original draft:** Jinxiong Chen.

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
