## [Decision Letter · Decision Letter 0]

3 Apr 2024

PONE-D-23-40567Research on the expansion, shrinkage characteristics and fracture evolution of red clay stabilised with phosphogypsum under dry-wet cyclesPLOS ONE

Dear Dr. Chen,

Thank you for submitting your manuscript to PLOS ONE. After careful consideration, we feel that it has merit but does not fully meet PLOS ONE’s publication criteria as it currently stands. Therefore, we invite you to submit a revised version of the manuscript that addresses the points raised during the review process.

Dear Authors,

The evaluations from the peer reviewers regarding your submitted work have been duly received. Upon reviewing their feedback, it is evident that they recommend that you revise your manuscript. Therefore, the authors should consider each comment and decide on the best course of action for their research.

We look forward to receiving your revised manuscript.

Kind regards,

Shaker Qaidi

Academic Editor

PLOS ONE

4. We suggest you thoroughly copyedit your manuscript for language usage, spelling, and grammar. If you do not know anyone who can help you do this, you may wish to consider employing a professional scientific editing service. 

A clean copy of the edited manuscript (uploaded as the new *manuscript* file)”.

 [Funding support: Supported by the Provincial Science and Technology Programme of Guizhou Province (Qiankehe Basic-ZK[2023] Key 016; Qiankehe Support[2020] 4Y038).].  

6. We note that your Data Availability Statement is currently as follows: [All relevant data are within the manuscript and its Supporting Information files.]

7. We note that Figure(s) 2, 13, 14 and 17 in your submission contain copyrighted images. All PLOS content is published under the Creative Commons Attribution License (CC BY 4.0), which means that the manuscript, images, and Supporting Information files will be freely available online, and any third party is permitted to access, download, copy, distribute, and use these materials in any way, even commercially, with proper attribution. For more information, see our copyright guidelines: http://journals.plos.org/plosone/s/licenses-and-copyright.

a. You may seek permission from the original copyright holder of Figure(s) 2, 13, 14 and 17 to publish the content specifically under the CC BY 4.0 license. 

8. Please include a copy of Table 8 which you refer to in your text on page 21.

Reviewers' comments:

Reviewer's Responses to Questions

**Comments to the Author**

1. Is the manuscript technically sound, and do the data support the conclusions?

Reviewer #1: Yes

Reviewer #2: Yes

Reviewer #3: Yes

2. Has the statistical analysis been performed appropriately and rigorously? 

Reviewer #1: N/A

Reviewer #2: N/A

Reviewer #3: Yes

3. Have the authors made all data underlying the findings in their manuscript fully available?

Reviewer #1: Yes

Reviewer #2: Yes

Reviewer #3: Yes

4. Is the manuscript presented in an intelligible fashion and written in standard English?

Reviewer #1: No

Reviewer #2: Yes

Reviewer #3: Yes

5. Review Comments to the Author

Reviewer #1: The manuscript used red clay and the waste of phosphogypsum in highway roadbed construction. The author analyzed the influence of phosphogypsum content on the performance of the mixture. The paper has certain engineering reference value. And the structure and methodology of the paper are both reasonable. However, some revisions are suggested based on the following comments.

1. The English grammar needs to be carefully checked and improved throughout the manuscript, e.g. by a professional English language editor. The current version is difficult for readers to comprehend.

2. The abstract needs to be rewritten, listing the main summary findings of this paper.

3. The keywords should be condensed further to accurately reflect the content of the paper.

4. In introduction section, in comparison to existing research, please explain the research objectives and innovations of the paper.

5. In the test section, the flowchart may be included to clearly demonstrate the research scheme of the paper.

6. The author emphasizes "special performance", but I believe the author only studied conventional engineering performance of the mixture of red clay and phosphogypsum. I suggest the author revise it.

7. In Analysis of test results section, some titles require modification to better align with the content discussed, for example, Relationship to compaction, Relationship with cement dosage, etc. are difficult to understand.

8. To enhance the theoretical depth of the paper, I suggest adding some mechanism analysis in the discussion section of the experimental results.

Reviewer #2: This manuscript investigates the expansion, shrinkage, and fracture evolution characteristics of red clay stabilized with phosphogypsum under dry-wet cycles. The authors conducted a thorough experimental study, varying key parameters such as cement dosage, compaction degree, water content, and number of dry-wet cycles. The results show that absolute expansion and shrinkage rates are positively correlated with compaction degree, number of dry-wet cycles, and cement dosage, while negatively correlated with initial moisture content and phosphogypsum dosage. Fracture rate increases with dry-wet cycles and decreases with initial water content, compaction, cement, and phosphogypsum dosage. The authors propose optimal cement and phosphogypsum dosages for roadbed filler. The research methodology is sound, and the findings have practical implications for road engineering. Some areas for improvement include providing more context, clarifying certain methodological details, and discussing limitations.

Comments:

1. The introduction could benefit from more context about the significance and challenges of using red clay as a roadbed material. What specific problems does the high liquid limit and plasticity of red clay pose for road construction?

2. On page 3, the authors mention that phosphogypsum contains toxic elements like arsenic, chromium, and lead. Have the environmental and health risks of using phosphogypsum been adequately addressed?

3. The sentence "From the above, it can be seen that there are certain research results on the strength and deformation of phosphogypsum soils, but there is a lack of research results related to the road performance of phosphogypsum-red clay due to the special characteristics of red clay" on page 3 could be supported with more specific examples of the research gaps.

4. How were the cement dosages of 4%, 6%, and 8% selected? Is there a reason these particular values were chosen?

5. The authors state on page 4 that the radioactivity index of the phosphogypsum is in accordance with GB 6566-2010 standards. It would be helpful to provide the actual radioactivity index values and limits for context.

6. Why was the optimal number of dry-wet cycles proposed to be 7 in this study (page 4)? Is this based on previous research or practical considerations?

7. On page 6, the humidification and drying target moisture contents are mentioned. How were these specific values (optimal +/- 7%) determined?

8. The image processing methodology using PCAS to quantify crack areas is interesting. Have the authors validated this technique against other methods or manual measurements?

9. In Section 3.1, the authors attribute the increase in absolute expansion rate with higher compaction to the increase in hydrophilic minerals. This explanation could be strengthened with more specific evidence or references.

10. The discussion on page 10 about closed pores transforming into connected pores and changes in the microscopic stress-strain field is intriguing. Is there any microstructural analysis that could support these claims?

11. On page 11, it's mentioned that CaO in cement reacts with SiO2 in clay to form hydrated calcium silicate gel. A reference for this reaction would be beneficial.

12. The explanation of phosphogypsum's inhibiting effect on the swelling and deformation of red clay (page 11) could be more detailed. What specific mechanisms are at play?

13. In Section 3.3, the authors discuss the development of horizontal tensile stress in the specimens during drying. Have the authors considered measuring or modeling these stresses directly?

14. On page 14, the authors state that an increase in cement dosage leads to more formation of hydrated calcium silicate and calcium aluminate, which increases tensile strength. This is a key point that warrants more discussion and supportive references.

15. The observation on page 15 that higher initial water content leads to the wrapping of micropores by water film and formation of a monolithic lamellar structure is noteworthy. Are there any microscopic images or previous studies that demonstrate this phenomenon?

16. The fitting equation (4) relating compaction, absolute expansion/shrinkage rate, and fracture rate is a valuable contribution. Have the authors considered the physical meaning behind the coefficients a, b, c, d, and e?

17. In the conclusion, the authors suggest an optimal cement dosage of 6% and phosphogypsum:red clay ratio of 1:1 to 1:2. How do these recommendations compare with current practice or other stabilization methods?

18. The research focuses on the experimental characterization of phosphogypsum-stabilized red clay. Are there any plans for field trials or case studies to validate the findings in real-world conditions?

19. The authors have demonstrated the potential of phosphogypsum for stabilizing red clay. Are there any economic or logistical considerations that could impact the widespread adoption of this technique?

20. While the study provides valuable insights, there are some limitations that could be addressed in future work. For example, the long-term durability and environmental impacts of phosphogypsum-stabilized red clay could be investigated further.

Reviewer #3: Please find the following the comments/suggestion.

1. The presented abstract is very length, it should be to the point and concise. Abstract could be more informative by providing results. I prefer to see some results in the abstract.

2. Please include the latest reference in the section 1. The introduction needs to be more emphasized on the research work with a detailed explanation of the whole process considering past, present and future scope. Please discuss more about the application of image processing in the other branches of engineering.

3. How the present study gives more accurate results than previous studies about the image processing? It needs to be strengthened in terms of recent research in this area with possible research gaps. It is strongly recommended to add a recent literature.

4. Research gaps should be highlighted more clearly and future applications of this study should be added.

5. There is no presentation of figures against the results, please include some graphs for the better understanding of the results.

6. Author use different abbreviation at different places, which confused the reader, Please provide the list of the abbreviation, please use in the start.

7. The manuscript required the proof reading.

8. Please pay attention on the formatting guidelines as per the journal requirement.

9. Please provide more discussion for the section 2, please include more description of the propose method by adding latest references.

10. In the conclusion section, the limitations of this study, suggested improvements of this work, and future directions should be added

The author needs to address the abovementioned points for the betterment of the manuscript.

6. PLOS authors have the option to publish the peer review history of their article (what does this mean?). If published, this will include your full peer review and any attached files.

Reviewer #1: No

Reviewer #2: No

Reviewer #3: No

---

## [Author Response · Author response to Decision Letter 0]

30 Apr 2024

Dear Academic Editors and Reviewers:

Thank you for your comments on the manuscript entitled “Research on the expansion, shrinkage properties and fracture evolution of red clay stabilised with phosphogypsum under dry-wet cycles”. The comments are very valuable and helpful for revising and improving our paper. We have studied the comments carefully and made correction accordingly. The main corrections and responses to the comments are listed as follows:（A more detailed list of responses is attached as “List of responses”）

Response to Academic Editorial Editor:

1. When submitting your revision, we need you to address these additional requirements. Please ensure that your manuscript meets PLOS ONE's style requirements, including those for file naming.

Response:

We thank the editors for their comments. We have reorganized the paper according to the PLOS One style.

Response:

We thank the editors for their comments. We have related them in the Methods section, and the experiments performed in this paper were conducted with permission from the laboratory, which is the Experimental Center of the School of Civil Engineering, Guizhou University, Guizhou, China.

Response:

We thank the editors for their comments. We have submitted the raw data from the trial as an attachment, which will be subsequently deposited in the repository.

4. We suggest you thoroughly copyedit your manuscript for language usage, spelling, and grammar. If you do not know anyone who can help you do this, you may wish to consider employing a professional scientific editing service.

Response:

We thank the editors for their comments, which were helpful in improving the quality of the paper, and our comments checked and corrected the language usage, spelling, and grammar of the manuscript.

 [Funding support: Supported by the Provincial Science and Technology Programme of Guizhou Province (Qiankehe Basic-ZK[2023] Key 016; Qiankehe Support[2020] 4Y038).]. 

Response:

We thank the editors for their comments. Our comments state in the cover letter the role the funder played in the study. The funder was not involved in study design, data collection and analysis, publication decisions, or manuscript preparation.

6. We note that your Data Availability Statement is currently as follows: [All relevant data are within the manuscript and its Supporting Information files.]

Response:

We thank the editors for their comments. We have submitted the raw data in a file called “Minimum Data Set” for your review.

7. We note that Figure(s) 2, 13, 14 and 17 in your submission contain copyrighted images. All PLOS content is published under the Creative Commons Attribution License (CC BY 4.0), which means that the manuscript, images, and Supporting Information files will be freely available online, and any third party is permitted to access, download, copy, distribute, and use these materials in any way, even commercially, with proper attribution. For more information, see our copyright guidelines: http://journals.plos.org/plosone/s/licenses-and-copyright.

a. You may seek permission from the original copyright holder of Figure(s) 2, 13, 14 and 17 to publish the content specifically under the CC BY 4.0 license. 

Response:

We thank the editor for his comments. The authors have replaced Figures 2, 13, and 14 referred to by the editor and removed Figure 17 from the manuscript. Since a flowchart has been added to the revised manuscript, the numbering of Figs. 2, 13, and 14 has been changed to Figs. 3, 14, and 15, respectively. in the new manuscript:

Fig. 3(a) shows the actual samples taken during testing, Fig. 3(b) is obtained by cropping Fig. 3(a), and Figs. 3(c) and (d) are obtained by software processing of Fig. 3(b);

All pictures in Figures 14 and 15 are cropped from the actual pictures taken during the test.

8. Please include a copy of Table 8 which you refer to in your text on page 21.

We thank the reviewers for their comments. Table 8 follows, and a copy of Table 8 has been submitted with the file name “Table 8”.

Response to Reviewer #1:

1. The English grammar needs to be carefully checked and improved throughout the manuscript, e.g. by a professional English language editor. The current version is difficult for readers to comprehend.

Response:

We thank the reviewers for their comments. The grammar of the manuscript has been checked and improved based on your comments.

2. The abstract needs to be rewritten, listing the main summary findings of this paper.

Response:

We thank the reviewers for their comments. The abstract of the paper has been rewritten based on your comments.

3. The keywords should be condensed further to accurately reflect the content of the paper.

Response:

We thank the reviewers for their comments. The keywords of the paper have been further streamlined based on your comments.

4. In introduction section, in comparison to existing research, please explain the research objectives and innovations of the paper.

Response:

We thank the reviewers for their comments. This suggestion is very helpful to us, and the revision has been completed according to the suggestion.

The purpose of this paper is to obtain the expansion and contraction properties and the expansion law of cracks of phosphogypsum-stabilized red clay under the action of dry and wet cycle by means of indoor test and image processing system, to provide theoretical basis for the application of phosphogypsum-stabilized red clay material in road engineering, and at the same time to improve the engineering characteristics of red clay. Innovations: (1) Innovation of research ideas: previous research is limited to the phosphogypsum mixed with other materials (soil, gravel, sand) to use, play an auxiliary role rather than as the main part of the use of this research according to the quality of phosphogypsum: soil = 1:1~1:5 to study. (2) Conclusion of the research innovation: phosphogypsum on the expansion and contraction of the mixture as well as cracking has a significant inhibitory effect. Phosphogypsum:red clay=1:1~1:2 as roadbed filler, greatly improve the utilization rate of phosphogypsum, but also can inhibit the red clay expansion and contraction and cracking.

5. In the test section, the flowchart may be included to clearly demonstrate the research scheme of the paper.

Response:

We thank the reviewers for their comments. Your comments were very helpful to our paper and we have followed the suggestion to add flowcharts to describe the experimental process.

6. The author emphasizes "special performance", but I believe the author only studied conventional engineering performance of the mixture of red clay and phosphogypsum. I suggest the author revise it.

Response:

We thank the reviewers for their comments. We have changed “characterization” to “properties” in response to the comments.

7. In Analysis of test results section, some titles require modification to better align with the content discussed, for example, Relationship to compaction, Relationship with cement dosage, etc. are difficult to understand.

Response:

We thank the reviewers for their comments, and we have revised the corresponding titles accordingly.

8. To enhance the theoretical depth of the paper, I suggest adding some mechanism analysis in the discussion section of the experimental results.

Response:

We thank the reviewers for their comments, which have been very helpful in improving the quality of our paper, and we have added the corresponding mechanistic analysis in the corresponding Discussion of Experimental Results section.

Response to Reviewer #2:

1. The introduction could benefit from more context about the significance and challenges of using red clay as a roadbed material. What specific problems does the high liquid limit and plasticity of red clay pose for road construction?

Response:

We thank the reviewers for their comments, your comments are valuable for our paper and we have followed the comments by presenting more space in the introduction section on the significance and challenges of red clay as a road base material, the specific problems caused by high liquid limit and high plasticity index for road construction such as instability of the road base, inhomogeneous settlement, drying and cracking, strength deterioration, poor drainage of the road base, and so on.

2. On page 3, the authors mention that phosphogypsum contains toxic elements like arsenic, chromium, and lead. Have the environmental and health risks of using phosphogypsum been adequately addressed?

Response:

We thank the reviewers for their comments. According to the test results in Table 4 of the paper, the heavy metal content of phosphogypsum complies with the relevant provisions of the national standard “Hazardous Waste Identification Standard Leaching Toxicity Identification” (GB5085.3-2007), and the radioactivity complies with the relevant provisions of GB6566-2010 “Radionuclide Limit for Building Materials”, so phosphogypsum will not have obvious impact on the environment and health when it is used in building materials.

Tab4. Test Results of Heavy Metals and Radioactivity in Phosphogypsum

Test items Standard limits Result Conclusion

Heavy metal

 Cu/mg‧L-1 ≤100 0.157 Qualified

 Zn/mg‧L-1 ≤100 0.051 Qualified

 Cd/mg‧L-1 ≤1 0 Qualified

 Pb/mg‧L-1 ≤5 0 Qualified

 Cr/mg‧L-1 ≤15 0 Qualified

 As/mg‧L-1 ≤5 0.0356 Qualified

 Hg/mg‧L-1 ≤0.1 0.0005 Qualified

Radioactivity Ra-226/Bq‧kg-1 — 53.94 —

 TH-232/Bq‧kg-1 — 42.13 —

 K-40/Bq‧kg-1 — 52.95 —

 IRa ≤1.0 0.3 Qualified

 Iγ ≤1.0 0.3 Qualified

3. The sentence "From the above, it can be seen that there are certain research results on the strength and deformation of phosphogypsum soils, but there is a lack of research results related to the road performance of phosphogypsum-red clay due to the special characteristics of red clay" on page 3 could be supported with more specific examples of the research gaps.

Response:

We thank the reviewers for their comments. It has been supported by using research examples based on the comments. From the literature [14-18], there are some research results on the strength and deformation of phosphogypsum stabilized soils, but the previous studies were limited to the use of phosphogypsum mixed with other materials (soil, gravel, sand), which played an auxiliary role rather than being used as the main part of the use of phosphogypsum, the phosphogypsum mixing is low, and due to the specificity of the red clay soil (high liquid limit, high pore ratio, and high water content), very few scholars have studied the road performance of phosphogypsum stabilized red clay soils. Due to the special characteristics of red clay (high liquid limit, high pore ratio and high water content), few scholars have studied the road performance of gypsum stabilized red clay.

14 ZONG Wei, WANG Yuanhui, XU Liang, LIU Cheng, ZHENG Wuxi. Research on road performance of industrial solid waste phosphogypsum pavement base material[J]. Silicate Bulletin,2024,43(02):766-773.DOI:10.16552/j.cnki.issn1001-1625.2024.02.006.

15 Luo Guofu, Chen Kaisheng, Luo Dipu. Compression characteristics and micro-mechanism of phosphogypsum-stabilised red clay[J]. Silicate Bulletin,2023,42(02):644-656.DOI:10.16552/j.cnki.issn1001-1625.20230207.004.

16 Qi J, Zhu H, Zhou P, Wang X, Wang Z, Yang Y, et al. Application of phosphogypsum in soilization : a review. INTERNATIONAL JOURNAL OF ENVIRONMENTAL SCIENCE AND TECHNOLOGY, 2023. 20(9): p. 10449-10464.

17 ZHOU Mingkai,ZHANG Xiaoqiao,CHEN Xiao,LI Lingzhi,WANG Ying,Research on the performance of cement phosphogypsum stabilized gravel pavement base material[J]. Highway,2016,61(04):186-190.

18 Wang Lei. Research on dynamic characteristics of roadbed with phosphogypsum stabilised soil [D]. Guizhou University,2022.DOI:10.27047/d.cnki.ggudu.2021.002322.

4. How were the cement dosages of 4%, 6%, and 8% selected? Is there a reason these particular values were chosen?

Response:

We thank the reviewers for their comments that according to the literature [22], the unconfined compressive strength of cement-phosphogypsum-red clay is greater at 4% to 8% cement dosage.

22 ZHANG Ying, CHEN Kaisheng. Study on the optimum dosage of red clay stabilized by phosphogypsum[J]. China Water Transportation(Next Half Month),2021,21(06):161-163.

5. The authors state on page 4 that the radioactivity index of the phosphogypsum is in accordance with GB 6566-2010 standards. It would be helpful to provide the actual radioactivity index values and limits for context.

Response:

We thank the reviewers for their comments. The radioactivity index values and limits have been listed in the t

---

## [Decision Letter · Decision Letter 1]

3 Jun 2024

PONE-D-23-40567R1Research on the expansion, shrinkage properties and fracture evolution of red clay stabilised with phosphogypsum under dry-wet cyclesPLOS ONE

Dear Dr. Chen,

After careful consideration, we feel that it has satisfied our scientific requirements for publication.

However, our editorial team have significant concerns about the grammar, usage, and overall readability of the manuscript. PLOS ONE requires that published manuscripts use language which is 'clear, correct, and unambiguous', see our criteria for publication at https://journals.plos.org/plosone/s/criteria-for-publication#loc-5. We therefore request that you revise the text to fix the grammatical errors and improve the overall readability of the text.

We suggest you have a fluent English-language speaker thoroughly copyedit your manuscript for language usage, spelling, and grammar. If you do not know anyone who can do this, you may wish to consider employing a professional scientific editing service.

Please note that we will not be able to proceed with publication of your manuscript until the concerns above are addressed.

* A copy of your manuscript showing your changes by either highlighting them or using track changes (uploaded as a supporting information file)

* A clean copy of the edited manuscript (uploaded as the new manuscript file)

We look forward to receiving your revised manuscript.

Kind regards,

Hanna Landenmark

Staff Editor, PLOS ONE

on behalf of 

Seyed Sina Mousavi Ojarestaghi

Journal Requirements:

Additional Editor Comments:

The authors appropriately enhanced the manuscript based on the reviewers' comments.

Reviewers' comments:

Reviewer's Responses to Questions

**Comments to the Author**

1. If the authors have adequately addressed your comments raised in a previous round of review and you feel that this manuscript is now acceptable for publication, you may indicate that here to bypass the “Comments to the Author” section, enter your conflict of interest statement in the “Confidential to Editor” section, and submit your "Accept" recommendation.

Reviewer #2: All comments have been addressed

2. Is the manuscript technically sound, and do the data support the conclusions?

Reviewer #2: Yes

3. Has the statistical analysis been performed appropriately and rigorously? 

Reviewer #2: Yes

4. Have the authors made all data underlying the findings in their manuscript fully available?

Reviewer #2: Yes

5. Is the manuscript presented in an intelligible fashion and written in standard English?

Reviewer #2: Yes

6. Review Comments to the Author

Reviewer #2: The manuscript (Research on the expansion, shrinkage properties and fracture evolution of red clay stabilised with phosphogypsum under dry-wet cycles) has been well revised and can be processed for the next stage of publication.

7. PLOS authors have the option to publish the peer review history of their article (what does this mean?). If published, this will include your full peer review and any attached files.

Reviewer #2: **Yes: **Mahmoud Akeed

---

## [Author Response · Author response to Decision Letter 1]

5 Jun 2024

Thank you very much for your valuable comments on the manuscript entitled “Research on the expansion, shrinkage properties and fracture evolution of red clay stabilised with phosphogypsum under dry-wet cycles”. In order to further meet the journal's publication criteria, we have scrutinized the linguistic presentation of the manuscript and completed the corresponding revisions, and the revised manuscript has been submitted to the journal.

---

## [Decision Letter · Decision Letter 2]

29 Jul 2024

Research on the expansion, shrinkage properties and fracture evolution of red clay stabilised with phosphogypsum under dry-wet cycles

PONE-D-23-40567R2

Dear Dr. Chen,

We’re pleased to inform you that your manuscript has been judged scientifically suitable for publication and will be formally accepted for publication once it meets all outstanding technical requirements.

Kind regards,

Jiaolong Ren

Academic Editor

PLOS ONE

Additional Editor Comments (optional):

Reviewers' comments:

Reviewer's Responses to Questions

**Comments to the Author**

1. If the authors have adequately addressed your comments raised in a previous round of review and you feel that this manuscript is now acceptable for publication, you may indicate that here to bypass the “Comments to the Author” section, enter your conflict of interest statement in the “Confidential to Editor” section, and submit your "Accept" recommendation.

Reviewer #2: All comments have been addressed

2. Is the manuscript technically sound, and do the data support the conclusions?

Reviewer #2: Yes

3. Has the statistical analysis been performed appropriately and rigorously? 

Reviewer #2: Yes

4. Have the authors made all data underlying the findings in their manuscript fully available?

Reviewer #2: Yes

5. Is the manuscript presented in an intelligible fashion and written in standard English?

Reviewer #2: Yes

6. Review Comments to the Author

Reviewer #2: Now, the manuscript (Research on the expansion, shrinkage properties and fracture evolution of red clay stabilised with phosphogypsum under dry-wet cycles) is ready for the next stage of the publishing process.

7. PLOS authors have the option to publish the peer review history of their article (what does this mean?). If published, this will include your full peer review and any attached files.

Reviewer #2: No

---

## [Editor Report · Acceptance letter]

12 Aug 2024

PONE-D-23-40567R2 

PLOS ONE

Dear Dr. Chen, 

I'm pleased to inform you that your manuscript has been deemed suitable for publication in PLOS ONE. Congratulations! Your manuscript is now being handed over to our production team.

Kind regards, 

on behalf of

Dr. Jiaolong Ren 

Academic Editor

PLOS ONE